# Pre-Clinical Evaluation of Efficacy and Safety of Human Limbus-Derived Stromal/Mesenchymal Stem Cells with and without Alginate Encapsulation for Future Clinical Applications

**DOI:** 10.3390/cells12060876

**Published:** 2023-03-11

**Authors:** Mukesh Damala, Abhishek Sahoo, Naveen Pakalapati, Vivek Singh, Sayan Basu

**Affiliations:** 1Centre for Ocular Regeneration (CORE), Prof Brien Holden Eye Research Centre, LV Prasad Eye Institute, Hyderabad 500034, India; 2School of Life Sciences, University of Hyderabad, Hyderabad 500046, India

**Keywords:** cornea, limbus, limbal stromal stem cells, stromal cell, immune response, toxicity, safety, cell encapsulation, efficacy, alginate, transport at room temperature

## Abstract

Corneal opacification or scarring is one of the leading causes of blindness worldwide. Human limbus-derived stromal/mesenchymal stem cells (hLMSCs) have the potential of clearing corneal scarring. In the current preclinical studies, we aimed to determine their ability to heal the scarred corneas, in a murine model of corneal scar, and examined their ocular and systemic toxicity after topical administration to rabbit eyes. The hLMSCs were derived from human donor corneas and were cultivated in a clean room facility in compliance with the current good manufacturing practices (cGMP). Before the administration, the hLMSCs were analyzed for their characteristic properties including immunostaining, and were further subjected to sterility and stability analysis. The corneas (right eye) of C57BL/6 mice (n = 56) were stripped of their central epithelium and superficial anterior stroma using a rotary burr (Alger Brush^®^ II). Few mice were left untreated (n = 8), while few (n = 24) were treated immediately with hLMSCs after debridement (prophylaxis group). The rest (n = 24, scar group) were allowed to develop corneal scarring for 2 weeks and then treated with hLMSCs. In both groups, the treatment modalities included encapsulated (En+) and non-encapsulated (En−) hLMSCs and sham (vehicle) treatment. The follow-up (4 weeks) after the treatment or debridement included clinical photography, fluorescein staining, and optical coherence tomography at regular intervals. All the images and scans were analyzed using ImageJ software to assess the changes in corneal haze, scar area, and the reflectivity ratio of the epithelium to the stroma. The scar area and the scar intensity were found to be decreased in the groups that received hLMSCs. The reflectivity of the stroma was found to be normalized to the baseline levels before the debridement in the eyes that were treated with hLMSCs, relative to the untreated. In the safety study, the central corneas of the left eye of 18 New Zealand rabbits were scraped with a needle and then treated with En+ hLMSCs, En− hLMSCs, and the sham (n = 6 each). Rabbits were then followed up for 4 weeks, during which blood and tear samples were collected at regular intervals. These rabbits were then assessed for changes in the quantities of inflammatory markers (TNF-α, IL-6, and IgE) in the sera and tears, changes in the ocular surface observations such as intraocular pressure (IOP), and the hematological and clinical chemistry parameters. Four weeks later, the rabbits were euthanized and examined histopathologically. No significant changes in conjunctival congestion, corneal clarity, or IOP were noticed during the ophthalmic examination. The level of inflammatory molecules (TNF-α and IL-6 TNF-α) and the hematological parameters were similar in all groups without any significant changes. Histological examination of the internal organs and ocular tissues did not reveal any abnormalities. The results of these studies summarize that the En+ and En− hLMSCs are not harmful to the recipient and potentially restore the transparency of debrided or scarred corneas, indicating that hLMSCs can be assessed for clinical use in humans.

## 1. Introduction

The cornea, also called the window of the eye, is a transparent structure in front of the eye. Light passes through it onto the retina, for the perception of light. Anatomically speaking, the ultrastructure of the cornea comprises three main layers: epithelium, followed by stroma, and followed by endothelium. The transparency of the cornea is due to the highly organized collagen fibrils in its stroma [1,2]. Corneal opacity or haze occurs when a person is exposed to infection, inflammation, or trauma [3,4]. Corneal scarring, a resultant of irregular fibrillogenesis following a wound, is one of the major causes affecting corneal transparency. Scarring involves the formation of atypical proteoglycans and the differentiation of the native keratocytes to the myofibroblastic phenotype [5,6,7,8].

The unavailability of standard treatments to clear corneal scarring makes corneal transplantation a pre-eminent mode of care for patients suffering from partial impairment of vision to complete blindness. The requirement for longer follow-up and the chances of graft rejection and the low rate of graft survival are the major limitations of corneal transplantation. Additionally, the unmet balance between the supply and the demand for donor corneas necessitates the need for alternative approaches to curb corneal scarring. Cell-based therapy is one of the emerging alternatives that could prevent and heal corneal scarring without the need for whole corneal transplantation [9,10,11].

Many groups across the globe have shown the potential of hLMSCs in preventing corneal haze [4,12,13,14,15,16]. Reports from the investigations by Basu et al. (2014) [4] and Du et al. (2009) [17] indicated that hLMSCs did not cause any immune reaction in the murine models of corneal scars. These cells are safe because it has been demonstrated that they can regulate the immune system [18] and that they do not produce any xenogeneic reactions in mouse models [4,18]. Various clinical studies are currently evaluating the safety and potency of hLMSCs and other mesenchymal stem cells [19,20,21,22,23,24]. By decreasing the need for donor corneas, the hLMSCs may reduce the need for corneal transplants. In addition, it has been demonstrated that hLMSCs preserved their viability and phenotype by being encapsulated in sodium alginate for 3–5 days while being transported or stored at various temperatures [25]. Without involving the patient in hundreds or thousands of kilometers of travel, this straightforward method, which does not require a costly cold chain, could expand access to hLMSC-based therapy, especially in rural and underdeveloped countries. However, before these novel techniques can be used in clinical settings, the toxicity and efficacy profiles of these cells, with or without encapsulation, must first be determined.

The objective of this study was to evaluate the cGMP-manufactured therapeutic-class hLMSCs for (a) harmlessness as well as detrimental effects following topical treatment in an animal model of corneal wound healing according to Indian regulatory guidelines, and (b) their effectiveness in preventing the formation of corneal scar and the regeneration of the corneal surface following treatment of the corneal scar with En−/En+ hLMSCs. Additionally, the information from the findings includes the vitality and stability of cGMP-grade hLMSCs throughout culturing and passages as well as the several quality checks that must be completed before these cells may be used in a clinical study.

## 2. Materials and Methods

### 2.1. Study Design and Ethical Approvals

#### 2.1.1. Approvals

The research ethics committee (Approval reference number 05-18-081) and the panel of the Institutional Committee for Stem Cell Research (Approval reference number ICSCR 08-18-002) at the LV Prasad Eye Institute, Hyderabad, approved the study methodology (Figure 1). The experimental protocols (safety study) on the animals were approved by the Animal Ethics Committee of Sipra Labs (Project reference number 110-19), Hyderabad, and adhered to the guidelines of Schedule–Y (26), Drugs and Cosmetics Rules act, 2019, Government of India (27).

The protocols used in the efficacy and safety studies were created in a way that complies with the ARVO Statement for the Use of Animals in Ophthalmic and Vision Research [26] issued by the Association for Research in Vision and Ophthalmology. All investigations conformed to generally accepted procedures, minimized or avoided the potential for animal suffering, and maintained their general health.

The International Council for Harmonization of Technical Requirements for Pharmaceuticals for Human Use (ICH) M3 (R2) [27] and the OECD (Organization for Economic Cooperation and Development) standards [28] of Good Laboratory Practice, 1997 were also followed in the conduct of this investigation.

#### 2.1.2. Donor Corneas

Therapeutic-grade donor corneas (n = 28) to harvest hLMSCs were obtained from Ramayamma International Eye Bank (RIEB), Hyderabad, India. The guidelines of the Declaration of Helsinki for the usage of human tissues were followed.

#### 2.1.3. Characterization of hLMSCs Expanded in GMP-Compliant Clean Room

All the batches of hLMSCs that were isolated and expanded using the optimized protocols underwent a series of tests and analyses (at both in-process and end-product stages) to ensure stability, sterility, and similitude of the characteristic properties. The tests included: qualitative and quantitative assessment of the phenotype through immunofluorescence and FACS (fluorescence-assisted cell sorting), karyotyping, quantification of the viability of hLMSCs in the cell pellet, microbial and mycoplasma analysis, determination of endotoxin content, and growth kinetics. The batches of hLMSCs that qualified for all of the tested parameters were used for the pre-clinical assessment in animal models.

#### 2.1.4. Assessment of the Efficacy of hLMSCs in a Murine Model of Corneal Scar

C57BL/6 mice (n = 56) of 6–8 weeks of age, weighing 20 to 25 g, were used for this study. A normal diet was provided. All mice were acclimatized to the cages at least a week before the beginning of the experimental procedures. The allocation was conducted through simple randomization. The mice were allocated to three study groups viz (a) the scar group (n = 24), (b) prophylaxis group (n = 24), and (c) untreated group (n = 8). The central epithelium and anterior stroma were debrided in the right eye of the mice. After debridement, the mice were treated either prophylactically (prophylaxis group) or therapeutically after allowing them to develop corneal scars for two weeks (scar group). Based on the method of treatment, these groups were divided into three subgroups, each based on the method of treatment: (i) sham (n = 8, vehicle only); (ii) En− hLMSCs (n = 8, cells that were neither encapsulated nor transported); and (iii) En+ hLMSCs (n = 8, cells released from transit after encapsulation). The untreated group was not provided with any treatment.

A clinical assessment of both eyes was undertaken before and after the debridement and treatment of the corneas. Clinical photographs of the ocular surface and optical coherence tomography (OCT) scans of the corneal ultrastructure were taken to detect the changes in the reflectivity and thickness of the corneal layers. Additionally, fluorescein staining of the ocular surface was performed to track the wound closure and reepithelization of the corneas debrided. The assessment was conducted at the stages of the pre-wound, wound, pre-op (on days 1, 7, and 14 during the development of the scar), and post-op stages (days 7, 14, 21, and 28).

#### 2.1.5. Determination of Safety and Toxicity of hLMSCs in Rabbits with Corneal Wounds

Three to four-month-old rabbits of the New Zealand White strain (n = 18) were used in this part of the study. The rabbits were allocated to the study groups through the stratified randomization method. Three groups of six rabbits each with three male and three female members received the following treatment: sham-treated group (G1) or control group; the G2 (En− hLMSCs) group received unencapsulated hLMSCs, while the G3 (En+ hLMSCs) group received encapsulated hLMSCs that were transported at room temperature.

The rabbits were anesthetized on the day of the experiment by injecting a formulation of ketamine (35 μg/g body weight) and xylazine (10 μg/g body weight). After that, 1–2 drops of topical anesthesia were applied to the eye (0.5 percent proparacaine). Next, a sterile needle was used to carefully scrape the corneal surfaces, as soon as they had been cleaned with a cotton swab soaked in 0.5 percent povidone-iodine. Then, the eyes of the G2 and G3 groups received 5 × 10^5^ En− hLMSCs and 50 × 10^5^ En+ hLMSCs, respectively, mixed with 100 uL of the fibrin glue formulation that is available for purchase (TISSEEL LYO, Baxter International Inc., Deerfield, IL, USA). The control group was treated with the vehicle alone (sham treatment) (i.e., fibrin glue) at the same time. To prevent the test item from being lost after the analytes were given, the eyes were closed for about 3 to 5 s. Finally, a sterile dressing pad was used to apply the treated eyes until the rabbits recovered from anesthesia. At each time point, additional ophthalmic examinations and blood analysis as well as the collection of serum and tear fluid were carried out. After the animals had been sacrificed, the pathological assessments were carried out on day 29.

### 2.2. Isolation and Expansion of hLMSCs

As previously reported [25], the limbal rim from the donor corneas served as the source of the hLMSCs. Briefly, limbal rims were dissected, cut to small fragments of 1–2 mm, and gently minced after the donor cornea was washed with the penicillin-streptomycin-gentamycin composition (15240062, Thermo Fisher Scientific, Waltham, MA, USA) diluted in PBS (14190250, Thermo Fisher Scientific, Waltham, MA, USA). Using the enzyme collagenase-IV (17104019, Thermo Fisher Scientific, Waltham, MA, USA), the minced limbal fragments were digested. After the digested tissue was washed, it was cultured in DMEM/F12 medium (BE04-687F/U1, Lonza, Basel, Switzerland) supplemented with 2% fetal bovine serum (SH30084.03, Cytiva Life Sciences, Marlborough, MA, USA). After reaching 80–90% confluence, the primary cultures (P0) were divided and subcultured for three generations or passages. At passage 3 (P3), a pure hLMSC culture was obtained, and post-viability checks were performed with 0.4% Trypan Blue (15250061, Thermo Fisher Scientific, Massachusetts, MA, USA).

A commercially available BeadReady^TM^ Kit [25] from Atelerix Ltd., Newcastle upon Tyne, TWR, UK was used to encapsulate hLMSCs with sodium-alginate. Using a sterile needle, the 2.5 × 10^6^ formulation of the alginate-cell suspension was released into a gelating buffer, where it polymerized into bead-like structures. For three to five days, these hLMSC-containing beads were in transit in a pre-standardized Styrofoam container that could maintain room temperature. These were suspended in the culture medium during the transit. The cells were then sedimented and released from the beads using a buffer containing trisodium citrate. For further analysis, the sedimented cell pellet was resuspended in a new complete medium. Before they were topically applied to the ocular surface, the pellet was washed with PBS/saline and the cell suspension was centrifuged at 1000 rpm for three minutes.

### 2.3. Analyzing the Distinctive Phenotype of hLMSCs

#### 2.3.1. Immunostaining

Until confluence, cells were cultured in 12-well culture plates with coverslips with a diameter of 18 mm at a density of 2 × 10^4^ cells per cm^2^ at 37 °C and 5% CO_2_. As described previously [25], hLMSCs were examined for the expression of typical markers of the MSC phenotype. The antibody panel featured markers for the human limbal stem cell trait such as Pax6, ABCG2, p63-α, and Col-III as well as markers for the MSC phenotype such as CD45, a negative indicator for mesenchymal cells, CD73, VIM, CD105, and CD90.

The minimum requirements for multipotent mesenchymal stromal cells as defined by the International Society for Cellular Therapy [29] were used to select this antibody panel. Alexa Fluor 594 (anti-mouse and anti-rabbit) from Thermo Fisher Scientific, Massachusetts, MA, USA, were included in the secondary antibody panel. A mounting medium (Fluoroshield, ab104139, Abcam, Cambridge, Cambs, UK) containing DAPI was used to mount the cells, and a Carl Zeiss Axio Scope A1 fluorescent microscope with a 20× or 40× objective was used for imaging. Biologic triplets were used in this experiment.

The number of viable cells was counted in a Neubauer chamber using the dye-exclusion method, which makes use of 0.4% Trypan Blue solution, and was used to measure the cell viability in both experimental groups. The minimum acceptance criterion was 70%, and the viability was expressed as percentage + SD.

#### 2.3.2. FACS

Fluorescence-assisted cell sorting (FACS) was used to quantitatively evaluate a portion of the populations of En−/En+ hLMSCs prior to administration to the murine corneas. After trypsinization from the cultures, viability checks were performed on both En− and En+ hLMSCs, and 10 uL of each primary antibody (diluted as per manufacturer’s instructions) was added to 50,000 En− hLMSCs in PBS after recovery from encapsulation, transport, and viability checks. The cells were then kept at 2–8 °C for 45–60 min in the dark. CD45, CD90, ABCG2, P63-α, and HLA-DR were the antibodies on the panel. As a control, no primary antibody was added to the cell suspension, so an “unstained” set of cells was used. After being incubated with the primary antibody, the cell suspensions were added to 200 mL of sheath fluid, and the CytoFLEX analyzer (Beckman Coulter, Indianapolis, IN, USA) was used for cytometric analysis.

### 2.4. Assessment of hLMSC Stability

#### 2.4.1. Evaluation of the Viability of Pelletized hLMSCs

The post-harvest cells from cultures and post-release cell suspensions after encapsulation (En−/En+ hLMSCs cell suspensions, respectively) were centrifuged at 1000 rpm for three minutes to eliminate the supernatant before being applied to the corneal surface. Since the process of transplanting the cell to the patient’s eyes usually takes some time, as does the journey from the GMP laboratory to the operation room, these pelleted cells were preserved at temperatures from 2 to 4 °C. To find the best window of time to transplant the cells onto the corneal surface, it is advised to evaluate their stability as a pellet. This was found by measuring the viability of these cells in pellet from the first hour to the end of 24 h from trypsinization. The cell suspension was evenly divided among six separate vials (0.5 × 10^6^ cells per vial/time point) and was preserved at temperatures from 2 to 4 °C following the initial viability evaluation. Using the dye-exclusion method, the amount (%) of viable cells at 30-min, 1-h, 3-h, 6-h, 12-h, and 24-h time points was measured and plotted.

#### 2.4.2. Karyotyping

A licensed third-party laboratory used karyotyping to look for chromatic defects and abnormalities in the hLMSCs. Colcemide was used to stop the spindle formation in hLMSC cultures that were three to four days old (with and without encapsulation). The chromosomes were then released from the cells by giving them a hypnotic treatment. After that, the G-banding method was used to prepare the slides, and a bright-field microscope was used to look at them. Cytovision^®^ software was used to carry out the analysis.

#### 2.4.3. Growth Kinetics

From the hour of seeding the cells to the completion of day 6 of expansion in the cell culture flask, the number of viable cells was measured using the MTT assay as well as the dye-exclusion methods. The doubling time and growth curve of the hLMSCs were obtained by plotting the data on a graph.

### 2.5. Assessment of the Sterility of hLMSCs

#### 2.5.1. Mycoplasma Assessment

Following the manufacturer’s directions when using the kit (LT07-318, MycoAlert^TM^, Lonza, Basel, Switzerland), the existence or absence of mycoplasma contamination was tested in the hLMSCs culture. A Luminometer (E5321, Promega, Wisconsin, WI, USA) was used to read the emitted light signal and check for mycoplasma in the cells’ spent media at the end of each passage and passage 3.

#### 2.5.2. Endotoxin Levels

A gel clot-based technique (N283-125, Lonza, Basel, Switzerland) was used to measure the amount of bacterial endotoxins (BET) existing in the hLMSC-suspension in conformity with the manufacturer’s protocol. The FDA’s rules [30] state that endotoxins cannot be present in amounts of more than 0.2 EU/mL.

### 2.6. Generation of the Murine Model of Corneal Scar

In normal saline, a mixture of xylazine and ketamine was used to anesthetize the mice. The mice were given 100 mg of xylazine (ilium Xylazil-100, Troy Laboratories Australia Pty. Ltd., NSW, Glendenning, Australia) and 10 mg of ketamine (Aneket^®^, Neon Laboratories Limited, Mumbai, India) per kilogram of body weight. Intraperitoneal administration of general anesthesia was conducted. Tearsplus (Allergan, Bangalore, India) lubricating eye drops were given to both eyes to keep them from drying out during the experiments. A surgical spear (EYETEC, Gujarat, India) was used to remove any objects or particles from the eyes, and the eyes were lubricated once more. After that, 0.5% proparacaine (Paracain, Sunways India Pvt Ltd., Mumbai, India) was applied topically to anesthetize both eyes.

Algerbrush^®^ II (Accutome Inc., Pennsylvania, PA, USA) with a 0.5 mm burr was used to gently rotate the right eye’s central cornea in a circular motion for 15–20 s. This removed the epithelium and a portion of the anterior stroma in the central cornea. The damage only affected the central cornea, not the limbus, sclera, or any other ocular surface area. The mice were either treated immediately or allowed to grow the scar for two weeks. After being gently scraped with a #15 surgical blade to remove the damaged tissue, the scarred or debrided corneas were treated with 5 × 10^4^ En−/En+ hLMSCs mixed in 2 μL of fibrin glue. Within one minute of application, this fibrin glue hardened into a gel-like clot. In each group, the contralateral eye (left) served as the normal control.

### 2.7. Assessment of Safety and Toxicity of hLMSCs

#### 2.7.1. Rabbit Body Weights and Death Rates

Every rabbit was checked for morbidity and demise twice daily. Additionally, on the first day of treatment and then every week after, the specific body weights (kg) were measured.

#### 2.7.2. Ophthalmic Investigations

The cornea, conjunctiva, iris, and aqueous humor were all examined using slit lamps (PSLAIA-11, Appasamy Associates, Chennai, TN, India). For corneal and conjunctival ophthalmic examinations, fluorescein ophthalmic strips were utilized. The ophthalmic observations were rated utilizing a numerical scoring procedure outlined in the OECD chemical testing guidelines, Test 405 “Scoring of the Lesions on Ocular surface” [28] and in accordance with Schedule Y [31] Before dosing, slit lamp and IOP readings were taken as well as at 3, 6, 12, and 24 h on day 1, and on days 7, 14, 21, and 28 after dosing. Appendix A outlines the scoring guidelines.

#### 2.7.3. Inflammatory Marker Quantification

At the end of 1, 6, 12, and 24 h on the day of treatment as well as on the days 7, 14, 21, and 28, blood samples ranging from 3 to 4 mL were taken from each animal using standard vacutainers. The blood samples were used to separate the sera, which was stored at –80 °C. Tear strips were used to collect samples of tear fluid at 1, 3, 6, 12, and 24 h as well as on days 7, 14, 21, and 28. For the purpose of determining the expression of the IL-6, TNF-α, and IgE markers, the collected samples were stored at –80 °C.

##### Schirmer Strip Tear Fluid Extraction

Applying the methodology that Posa et al. had previously published [32], Schirmer’s strip (Tear Strip, Care Group, Vadodara, GJ, India) was used to extract the tears. Using forceps, the frozen strips were inserted into a sterile 0.5 mL microcentrifuge tube. A fresh 22 ½ gauze needle was used to puncture these microcentrifuge tubes containing 0.5 mL. A 1.5 mL microcentrifuge tube was used to store the entire arrangement. Next, 10–50 mL of 1× PBS was added to the strip, based on the strip length in millimeters. The strip was then incubated for 30 min at 2–4 °C. Afterward, the apparatus was centrifuged at 4 °C for 5 min at 13,000 rpm. Each microliter of the collected tear fluid was evaluated to determine the level of protein, with the remaining volume being subsequently frozen at −80 °C for further study.

##### BCA Protein Quantitation

In accordance with the manufacturer’s instructions, the bicinchoninic acid (BCA) assay (786-570, G-Biosciences, Geno Technology Inc., St. Louis, MO, USA) was used to measure the amount of protein in the tear samples collected. The standard graph obtained was compared to the concentration of the unknown samples. Using a SpectraMax M3 spectrophotometer (Molecular Devices, San Jose, CA, USA), the absorbance was measured at 562 nm for the standards, which ranged from 2000 pg to 0 μg/mL.

##### Quantification of Markers through Immunoassay

Using sandwich ELISA, the levels of inflammatory markers in rabbits were measured. KinesisDx, Krishgen Biosystems, USA, supplied commercially available antibody-coated kits for the quantification (IgE, K09-0071; IL-6, Ref: KLX0065), TNF-α, and KLX0003. Briefly, 10 μL of each biotinylated antibody was added to each well after 40 μL of each sample (sera/tear) was added. There were no biotinylated antibodies in the standards. The streptavidin-HRP conjugate solution was then added to all wells and stored in an incubator at 37 °C for one hour in the dark. The wells were then thoroughly tapped onto absorbent paper and washed four times with washing buffer by utilizing an automatic washer (Erba Lisa Wash II, Erba Mannheim, Brentford, LDN, UK). After that, 50 μL of substrate A, 50 μL of substrate B, were added to the wells and incubated for 10 min. The SpectraMax M3 spectrophotometer was used to read the formed color at 450 nm following the addition of 50 uL of stop solution per well to halt the reaction.

#### 2.7.4. Blood Investigations

Using a hematology cell quantifier (SYSMEX-XP 100, Kobe, OC, Japan), the hematological parameters were determined. The Leishman stain was used to stain the hematology sample to make blood smears. Utilizing standard microscopy, for these smears, the differential leukocyte count was performed. Clinical chemistry analysis was performed on the sera that were extracted from the blood specimens. A fully automated Random Access Biochemical Analyzer was used to perform the clinical chemistry test (EM-360, Erba Mannheim, Brentford, LDN, UK).

#### 2.7.5. Tissue Evaluations

##### External Examinations and Necropsies

After the study duration, every single rabbit was sacrificed and underwent a thorough necropsy. The gross findings that might point to abnormalities were noted. During an in situ examination, the individual organs were investigated for histomorphological anomalies.

##### Organs Weights and Histopathology

The organs were collected and weighed after the gross pathology examination was finished. The ratios of organ weight to body weight were calculated. For histopathological examination, 10% buffered formalin preserved the organs.

### 2.8. Statistical Analysis

Mean + SD was used to represent all of the data. Using GraphPad software, the findings were all put through statistical analysis with a significance level (of 0.05). The Student’s *t*-test (safety study—organ and body weights, clinical and hematological parameters) and non-parametric one-way ANOVA (Kruskal–Wallis) tests (safety study—IOP, inflammatory marker assessment; efficacy study—changes scar intensity, scar area, and E:S ratios) were used to analyze the data.

## 3. Results

### 3.1. Characteristic Analysis of hLMSCs

#### 3.1.1. Phenotypic Assessment of hLMSCs

Col-III, p63-α, Pax6, and ABCG2 were both expressed positively by the cells. As expected, mesenchymal biomarkers such as CD73, VIM, CD73, CD105, and CD90 were expressed positively, but CD45 was not. Overall, the phenotypic expression of the hLMSCs of the biomarkers was found to be unaltered (Figure 2A).

#### 3.1.2. Evaluation of the Viability and Stability of hLMSCs

Karyotyping revealed no numerical or chromatic aberrations in either of the En− or En+ hLMSC cell populations (Figure 2B). At the end of six hours, 88.33 ± 2.37% of the pelleted hLMSCs were still alive, while at the end of 24 h, 78.21 ± 1.47% of the cells were still alive (Figure 2C). The doubling time of hLMSCs was less than 61 h, according to the growth kinetics studies. In both of the En−/En+ hLMSCs that were administered to the study’s test animals, there was no evidence of Mycoplasma species contamination. The En− hLMSCs and En+ hLMSCs cell suspensions had levels of bacterial endotoxins that were within the acceptable range (<0.12 EU/mL).

### 3.2. Comparison of the Effectiveness of the hLMSCs with and without the Incorporation of Alginate

Debridement of the corneal epithelium and stroma successfully led to the formation of scarring or haze (Figure 3). The reepithelization of the cornea was observed to happen more or less in the first two weeks in all groups. Groups that received hLMSCs in both the scar and prophylaxis groups were found with similar levels of tissue regeneration and the restoration of the transparency in terms of the scar intensity (Figure 4 and Figure 5A,B,D,E).

#### 3.2.1. Change in Corneal Haze

In both the prophylaxis and scar groups, the intensity of the corneal scar or haze in the eyes that received En−/En+ hLMSCs decreased toward the conclusion of the investigation in comparison to the pre-treatment (*p* < 0.0001, n = 6). In mice that received En− and En+ hLMSCs, the intensity of corneal haze decreased from 164 ± 12 GSU and 164 ± 11 GSU on day 14 of scar formation to 121 ± 6 GSU and 124 ± 11 GSU at the end of day 28 (Figure 5A–C).

In a similar vein, the prophylaxis group’s haze decreased to 138 ± 19 and 136 ± 11 GSU on day 28 after treatment, as opposed to 151 ± 14 and 173 ± 13 GSU on day 1 of the wounding and transplantation of En+ hLMSCs, respectively (Figure 5B). In contrast, there was no significant change in the corneal scar intensity in the eyes that received the sham treatment or no treatment (Figure 5A,C) compared to the baseline prior to transplantation.

#### 3.2.2. Reduction in the Scar Area

The scarred corneal surface area gradually decreased in all treatment arms (Figure 5D,E) after En− and En+ hLMSC treatment, over the course of the two-week scar development period. From day 7 of scar development to day 28 of treatment, the mice that received the sham treatment maintained corneal scarring of the same size (Figure 5D,E).

By the end of the study, the eyes that received En−/En+ hLMSCs immediately after debridement had a level of scarred corneal surface that was consistent, with a slight decrease in the area that was statistically insignificant (*p* = 0.0875). On the other hand, similar to the scar (*p* < 0.001, Figure 5D) and untreated (*p* < 0.0001, Figure 5F) groups, the eyes that received the sham treatment prophylactically displayed an increase in the scarred area that remained unchanged throughout the follow-up.

#### 3.2.3. Epithelium to Stroma Reflectivity

Before any wound was made, the average E:S reflectivity ratio of the three groups ranged from 0.87 ± 0.03 to 0.96 ± 0.01.

While the eyes that received En−/En+ hLMSCs were able to normalize to the baseline E:S ratio in all of the treated arms (Figure 5G,H), this ratio was found to gradually de-crease in the untreated (0.96 ± 0.01 to 0.65 ± 0.02) or sham-treated arms (scar: 0.93 ± 0.04 to 0.68 ± 01 and prophylaxis: 0.96 ± 0.01 to 0.76 ± 0.1) in all of the groups, indicating the elevated stromal reflectivity (Figure 5I).

### 3.3. Determination of the Safety and Toxicity of hLMSCs

#### 3.3.1. Clinical Symptoms, Body Weights, and Death Rate

All of the animals in the sham and test (En+/En− hLMSC) groups showed no clinical signs. In both the sham and test groups, there was no mortality. When compared to the control group, a normal weight increase was determined to have occurred in all of the test groups (Appendix A).

#### 3.3.2. Ophthalmic Observations and IOP

It was found that all of the ophthalmic findings were normal. However, at the three-hour time point, the left conjunctivas of all three groups showed Grade 1 ocular inflammation. At the 6 h time point, the same happened to one of the six sham group animals and to all animals in the En− hLMSC group. From the 12th hour onward, there were no symptoms of ocular irritation observed. In all three groups, the contralateral (normal) eyes did not exhibit any ocular lesions and remained normal at all time points throughout the study (Figure 6 and Appendix A).

In all three groups, the intraocular pressure was found to be comparable within the normal range. The IOP of the treated eyes in either the test group or the control group was not significantly dissimilar from the sham or control group. In all groups, the IOP of the opposite eye (normal) also did not change significantly, with the exception of a single time point, day 28 (Figure 6 and Appendix A).

#### 3.3.3. Evaluation of Immunogenicity and Inflammatory Markers

The rabbit sera showed that the inflammatory markers TNF-α and IL-6 declined. In both test groups (En+/En− hLMSCs), the mean concentrations of these analytes were found to decrease in a manner that was comparable to that of the control group (G1) (Figure 7E,F). The outliers were a few occurrences in the very early stages (level of TNF-α in tears at hours 1 and 3 after treatment), and it was discovered that the TNF-α and IL-6 levels, two inflammatory chemicals, were pointedly low and also seen to decline throughout the study (Figure 6C and Figure 7B). At five of the eight time points, the serum IgE levels in the En+ hLMSC group were higher than those of the other two groups (Figure 7D). In contrast, except for the first and third hours of treatment for the En−hLMSC group, IgE levels in the tear samples were shown to decrease (Figure 7A). In general, all three groups maintained comparable levels of IgE in the tears.

#### 3.3.4. Hematology

The sham and test item transplanted groups (En+/En− hLMSCs) had similar hematological values (Appendix A). The bone marrow showed no hematopoietic system changes. In the sham/control group, no test group showed erythropoiesis, granulopoiesis, or lymphopoiesis. Appendix A shows that none of the G1, G2, or G3 animals had hypocellularity, hypercellularity, or hypochromatism.

One G1 and G2 rabbit produced granulopoietic cells. These modifications were absent in G3 (En+ hLMSCs) granulopoietic cells. The cells did not impact granulopoietic activity in comparison to the sham group. Some animals in the control and hLMSC transplanted groups showed changes in granulopoietic activity, indicating that their immune systems spontaneously changed.

Bone marrow smears taken from all of the animals in groups G1, G2, and G3 indicated that there was no toxicity or dose-dependent change in the synthesis of precursor cells for myeloid, erythroid, or lymphoid cells. This was the case in comparison to the “sham” or “control” group, which was given zero doses of the cells.

#### 3.3.5. Clinical Chemistry

Except for the following observations, all of the clinical chemistry values were found to be normal. When compared to the sham group, the levels of phosphorus in the G3 group were higher (7.35 ± 1.11 mg/dL) (5.83 ± 0.39 mg/dL). Total proteins decreased by 5.63 ± 0.38 g/dL, globulin decreased by 3.27 ± 0.21 g/dL, and sodium decreased by 153.26 ±5.01 mmol/L in the G2 group. When compared to the animals in the sham group, the level of sodium in the G3 group animals was lower (152.47 ± 1.86 mmol/L) (Appendix A). The internal organs of the animals were unaffected by the observed changes.

#### 3.3.6. Organ Weights, Gross Observations, and Necropsy

In all animal groups, both external and internal examinations of the organs revealed no abnormalities (Appendix A). In each of the G2 and G3 groups, the organ weights were found to be normal. When compared to the animals in the control group, the test item administered animals underwent no significant changes.

#### 3.3.7. Histopathology

Compared to the control group, the majority of organs did not show any abnormal findings or changes (Appendix A).

Two rabbits from each group—the G1, G2, and G3 groups—had sinusoidal hemorrhages in their livers [2 of 6]. One animal from the G1 group showed necroses and infiltration of inflammatory cells, but the livers of the other groups were unaffected. Five G1 and five G2 animals had alveolar thickening or inflammation.

One animal of the G2 group and the G3 group both had kidneys with tubular degeneration. All groups—G1 group [3 animals], G2 group [1 animal], and G3 group [1 animal]—were found to have foci of tubular or interstitial inflammation. Two G1 and one G2 animals had cerebral hemisphere necrosis, and G3 did not show brain alterations. One G1 male, one G2 male, and one G3 female developed submucosal lymphoid tissue hyperplasia in their ilium mucosa.

However, when compared to the G1 group, the ileum, lung, liver, kidney, eye, and kidneys showed no dose-related adverse effects. Since these organ lesions emerged in both the vehicle control group and the test item group, it is possible that they developed on their own. Additionally, there were no consistent or significant lesions in these organs between the vehicle control animals and animals given the test item. In conclusion, none of the systemic organs underwent significant reactive or toxic changes (Appendix A).

## 4. Discussion

In recent years, numerous potential treatments for corneal opacification and scarring other than corneal transplantation have emerged. Biomimetic hydrogels, cell-based methods, and molecular methods are examples of these. Different hydrogels—with or without cells—have been demonstrated in several studies to be an effective option for stromal replacement using donor tissue. [33,34,35,36]. Exosomes [37], anti-TGF- [6,7,38], anti-PDGF [7,39,40], and HGF [41,42] have all been shown to play a role in either preventing or reversing corneal scars. During wound healing, researchers have found that corneal scars can be repaired in two ways: by reversing the conversion of myofibroblasts to fibroblasts or by inhibiting TGF-/SMAD signaling [4,43,44,45,46]. In the past few years, hLMSCs have demonstrated promising latent for non-scarring wound healing from various pathologies [4]. When these cells are encased in alginate, it has also been demonstrated that they maintain their characteristic properties and have a longer shelf life when subjected to a variety of temperature conditions [25]. Without the need for costly cold-chain systems, alginate encapsulation can make it easier for these cells to travel over long distances. As stromal scarring or opacification-related corneal blindness prevalence is highest in developing nations, more affordable and simpler transportation will make patients in remote areas more accessible to cell-based treatments at lower costs. The aim of this study was to determine the toxicity of hLMSCs after they were applied topically to rabbit corneas and their potential in healing and preventing corneal scars in a murine model.

According to previously reported studies [25], the limbus-isolated LMSC donor corneas were cultivated in a CGMP-grade cell culture suite. After topical treatment on rabbit and mouse eyes with corneal lesions and scars, the efficacy and toxicity of LMSCs encapsulated in alginate and transited for three days and those not encapsulated were assessed. The vehicle served as a sham for the control group, which received no cells. After the treatment, clinical imaging and optical coherence tomography were used to examine the eyes of mice for a period of four weeks. Through ophthalmic, hematological, and tissue examinations of the rabbits, comprehensive evaluations of the toxicity to the system as well as the eyes were carried out. Throughout the course of the study, there was no mortality in the animals.

In groups that received therapeutic (scar group) or prophylactic (prophylaxis group) treatment for the murine eyes, the scarring was cleared or prevented. When compared to the sham-treated or untreated groups, these arms showed a decrease in corneal haze, or scar area and intensity.

At the conclusion of the safety study, all rabbits were sacrificed, and all major organs including the eyes were taken and examined histologically in detail. The intraocular pressure of the treated rabbits did not significantly change during the ophthalmic examinations (Figure 7 and Appendix A), which also revealed normal observations of IOP (Appendix A and Figure 7). In all three groups, the hematological examination parameters were comparable (Appendix A). Histopathological examination revealed no abnormalities in the corneal tissues (Figure 8). Against the sham group, neither the tears nor the sera of the experimental groups displayed any significant signs of an inflammatory response (TNF-α and IL-6) (Figure 7A–F). This study offers additional proof for the safety of hLMSCs, suggesting that human clinical trials may evaluate these cells for clinical applications.

Regenerative medicine’s recent advancements have made it possible to treat a wide range of diseases and disorders. One of the main therapies being tested in clinical trials around the world for their efficacy in treating heart, ear, bone, and eye diseases is mesenchymal stem cell therapy [47,48]. However, guaranteeing the patient’s safety is the most crucial element and the top concern of any clinical investigation or pharmaceutical development process. In order to determine the toxicity or safety profile of the drug or cell product, preclinical testing and compliance with various regulatory requirements are required. MSCs derived from bone marrow have been shown to be safe and effective for corneal repair in a recent study by Putra et al. [49]. The aforementioned study was carried out in advance of the Phase I clinical trial. However, the safety of GMP-manufactured human limbus-derived MSCs for upcoming clinical trials is poorly documented in the literature. The Drug Controller General of India, part of the Central Drugs Standards Control Organization (CDSCO), regulates India’s pharmaceuticals as the FDA does in the U.S. According to The Government of India’s Drugs and Cosmetics Rules, 2018 (Schedule Y) [31,50], these bodies require the safety evaluation of each drug and surgical procedure [31,50]. In this study, hLMSCs were evaluated in accordance with the above laws and the Good Laboratory Practice (GLP) guidelines by the OECD. It has been demonstrated that encapsulating corneal epithelium and hLMSCs in sodium alginate [25,51] may increase the cells’ shelf life, enabling room-temperature transport while maintaining their distinctive phenotype and vitality. With the potential to considerably reduce associated expenses, this technique significantly improves the costs of this new advanced cell-based therapy. Because it eliminates the time-consuming and costly cold-chain transport and has the potential to considerably reduce associated expenses, this technique significantly improves the finances of this new advanced cell-based therapy.

In the scar group that were treated after the scar developed, the cross-sections of murine corneas in the OCT scans revealed a significant reduction in the area affected by scarring (Figure 5D). In the group that received the En−/En+ hLMSCs prophylactically, there was also a decrease in the scar area (Figure 5E), but it was not statistically significant. However, all of the groups that received hLMSCs had significantly less corneal haze (Figure 5A,B). In terms of the scar area and intensity of the treated corneas, the groups that received sham or no treatment had comparable outcomes (Figure 5C,F). This clearly demonstrates that the hLMSCs assist in the repair of corneal wounds or scars.

In addition, during the same time period following treatment, the scar area had diminished to numbers that were comparable (ranging from 412 to 488 microns in the prophylaxis group and 501 to 512 microns in the scar groups). This degree of similarity in the scar area demonstrates that the hLMSCs are able to restore the damaged corneal surface without causing any scarring and heal corneal scars (scar group, treated two weeks after scar development). Additionally, it demonstrates that the alginate encapsulation has no effect on the efficacy of hLMSCs (Figure 5A,B,D,E).

In the arms of both the scar and prophylaxis groups that received hLMSCs, the corneal surface transparency returned to its pre-debridement readings (Figure 5G,H). The transparency of the cornea was impacted by the increased reflectivity of the stromal surface in the eyes of the untreated group (Figure 5I). The untreated groups’ corneal reflectivity increased by 32.3%, while the scar and prophylaxis groups’ reflectivity increased by 26.7 percent and 20.8 percent, respectively, in the sham-treated arms.

According to the rather small amounts of cytokine molecules IL-6 and TNF-α in the tears, the evaluation of inflammatory cytokines demonstrated that these cells did not cause eye toxicity (Figure 7B,C). Similar findings were made regarding the systemic toxicity of these cells from the levels of the analytes TNF-α (Figure 7F) and IL-6 (Figure 7E) in the rabbits’ blood serum. At specific time points, animals that were given cells released from transit had significantly higher levels of IgE molecules than the control/sham group and the group treated with non-encapsulated cells, indicating any potential allergens. However, neither the amounts of IgE in the tears of the hLMSC-treated animals nor the varying levels of IgE were accompanied by a clear trend (Figure 7D). The TNF-α and IL-6 expression in the tear samples were significantly lower in both experimental arms (Appendix A and Figure 7B,C). In addition, no ocular lesions were observed after 12 h post-treatment until the study’s conclusion, and eye examination proved to have insignificant variations in IOP levels (Figure 6 and Appendix A). According to the results of the histopathological examinations (Figure 8 and Appendix A), the variations that were observed in the clinical chemistry parameters (Appendix A) and hematological indicators (Appendix A) did not affect the systemic organs. In addition, the data in Figure 5B demonstrate the stability of the cells, sterility, and no chromosomal abnormalities support the safety of the cells for human testing.

The fact that this study was conducted at a GLP-certified animal facility with a NABL accreditation (National Accreditation Board for Testing and Calibration Laboratories) facility is a strength. Veterinarians, biochemists, and pathologists were all hidden from the intervention under investigation. Compared to the previous study [25], this one did not include the hLMSCs transiting for more than three days after alginate encapsulation, which may be a limitation. However, this time frame was chosen in light of the fact that the cells would be able to reach any faraway part of the country within three days of being distributed. It is possible that by evaluating tears from an untreated or healthy eye, ocular toxicity may have been better assessed. These LSMCs were solely applied to the corneal surface in this study, which is also the planned route of administration for the clinical trials. However, introducing these hLMSCs to the subconjunctival area might provide the possibility of investigating not just the various delivery mechanisms, but also their safety. This will be investigated in the future.

## 5. Conclusions

The purpose of our study was to establish the efficacy and toxicity of hLMSCs in wounded rabbits and murine corneas, and whether they were encapsulated in alginate or not. Our study suggests that the hLMSCs are safe because they do not harm the recipient and do not cause any inflammatory response. hLMSCs are able to repair traumatized tissues and effectively restore corneal surface transparency. This ensures that these cells can be used on humans to test their efficacy in treating corneal wound healing. In the end, this will make them more affordable and available to people in the most remote places, eliminating the need for long-distance travel.

## Figures and Tables

**Figure 1 cells-12-00876-f001:**
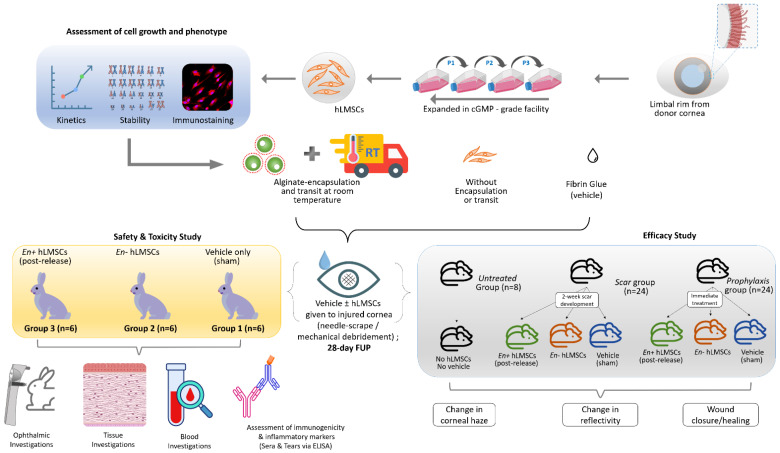
Graphical abstract of the experimental plan for assessing the safety and efficacy of human limbus−derived stromal/mesenchymal stem cells.

**Figure 2 cells-12-00876-f002:**
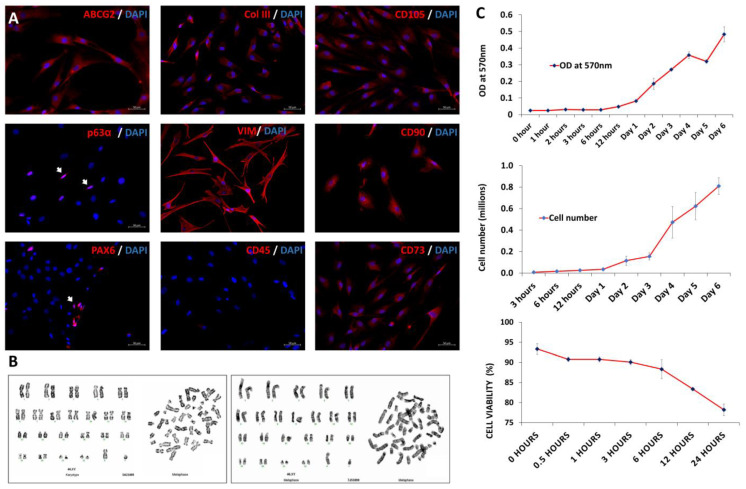
hLMSC phenotyping and stability. (**A**) Immunostaining assessment of the hLMSC phenotype before administering them to rabbit corneas. The panel shows stem-cell biomarkers (p63+, Pax6+, ABCG2+) and mesenchymal biomarkers (VIM+, CD45−, CD73+, CD90+, and CD105+) stained red against DAPI, nuclear stain (blue). 40×; 50 μM. (**B**) Karyotyping of hLMSCs before and after encapsulation and transport (n = 3). Both groups showed no numerical or significant reforms. (**C**) Top graph shows the hLMSC growth in culture. Third-passage cells were seeded in equal numbers into well plates and assessed via the MTT assay for 7 days (n = 3). The 570 nm absorbance was plotted against culture duration. The dye-exclusion graph of the hLMSC culture growth (middle). The bottom graph shows the viable cell percentage in pellet at various time points when stored at 2–8 °C. The hLMSCs were stable with 90.09 ± 0.06 percent viability at 3 h and 88.33 ± 2.37 percent viability at 6 h (n = 3), the timeframe for corneal transplantation.

**Figure 3 cells-12-00876-f003:**
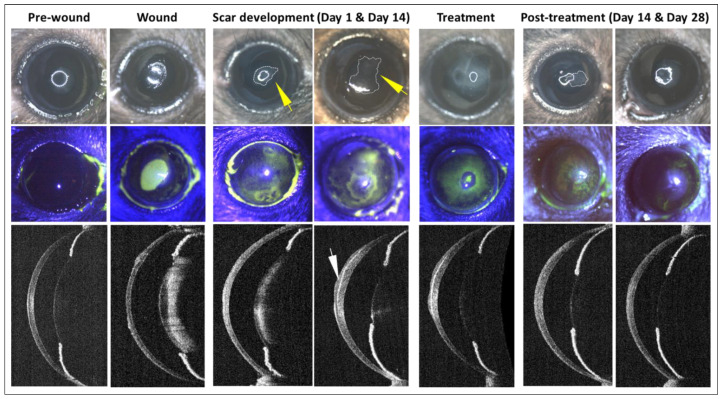
Generation of the corneal scar, treatment, and clinical follow-up. Collage of the representative (scar group) clinical photographs (top row) of the normal corneal surface (pre-wound) before debriding the central cornea (wound), and the respective fluorescein staining images confirming the compromised epithelial integrity (middle row). The debrided corneas developed clouding or haze (marked with a dotted line, indicated with a yellow arrow) on day 1 after the debridement and by the end of two weeks, the debrided area developed a scar. The respective OCT scan on the lower panel shows the scarring in the anterior stroma (indicated by a white arrow) and altered corneal thickness. The scarred tissue was scraped away and treated with hLMSCs (treatment) in fibrin glue. The OCT scan on day 28 post-treatment shows stabilized corneal transparency relative to the scarred sections.

**Figure 4 cells-12-00876-f004:**
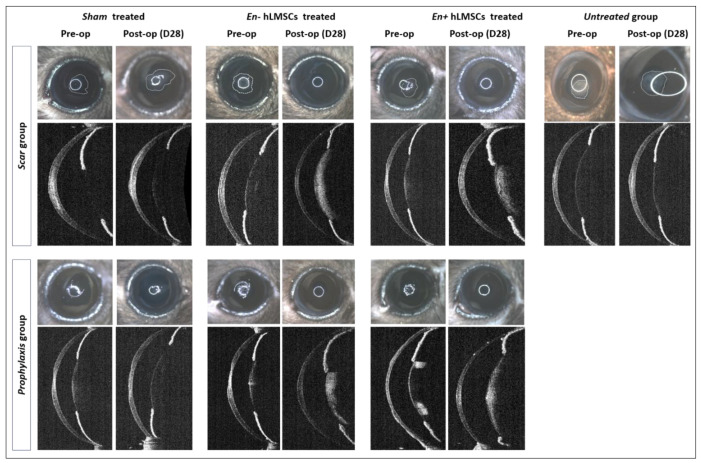
Outcomes of the treatment with hLMSCs: Collage of representative microphotographs and scans showing the scarring of debrided corneas before and after treatment with hLMSCs. Eyes of the untreated and sham treated arms show the unhealed corneas post debridement/treatment. Eyes treated with En−/En+ hLMSCs showed relatively clear corneas with less haze and scarring.

**Figure 5 cells-12-00876-f005:**
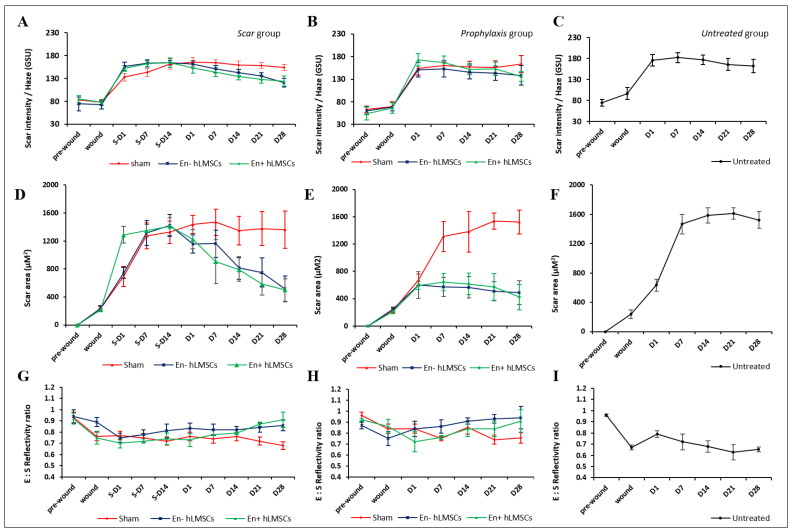
Changes in the corneal scar intensity, area, and reflectivity. (**A**–**C**) Graph plots showing the relative decrease in the corneal haze or the scar intensity of the murine eyes treated with encapsulated and non-encapsulated hLMSCs, and both. Corneas treated with vehicle alone (sham) or left untreated remained without any significant change up to the endpoint of the study. (**D**–**F**) Graph plots showing the reduction in the size of the corneal scars. Mice treated with hLMSCs after scar development (**D**) showed a significant decrease (*p* < 0.0001, n = 8) in the scar area, relative to pre-treatment (S-D1 to S-D14), whereas the mice that received hLMSCs prophylactically did not show any significant (*p* = 0.08, n = 8) increase in the scar area. (**G**–**I**) The reflectivity of the corneal surface normalized to the baseline readings in the eyes that received hLMSCs in both the scar (**G**) and prophylaxis (**H**) groups. The reflectivity of the stroma increased in eyes that received the sham (**G,H**) or no treatment (**I**).

**Figure 6 cells-12-00876-f006:**
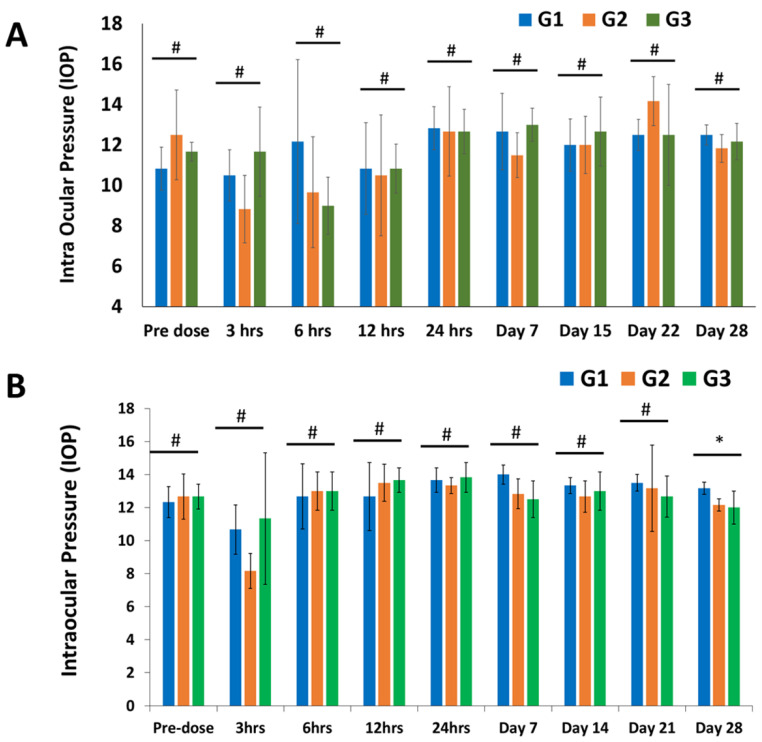
Changes in the intraocular pressure of the rabbits at different study time points. (**A**) Intraocular pressure (IOP) was monitored before and after treatment and is depicted as a bar graph. Comparing the experimental groups (G2 and G3) to the control group, no statistically significant differences were found in IOP levels (G1). n = 6; * *p* < 0.05, # *p* > 0.05. (**B**) Intraocular pressure (IOP) variations in healthy eyes, represented as a bar graph. Except for one time point, there were no statistically significant differences between the IOP levels of the experimental groups (G2 and G3) and the control group (G1) (Day 28). n = 6; * *p* < 0.05, # *p* > 0.05. G1—Sham treated group; G2—Treated with En− hLMSCs; G3—Treated with En+ hLMSCs.

**Figure 7 cells-12-00876-f007:**
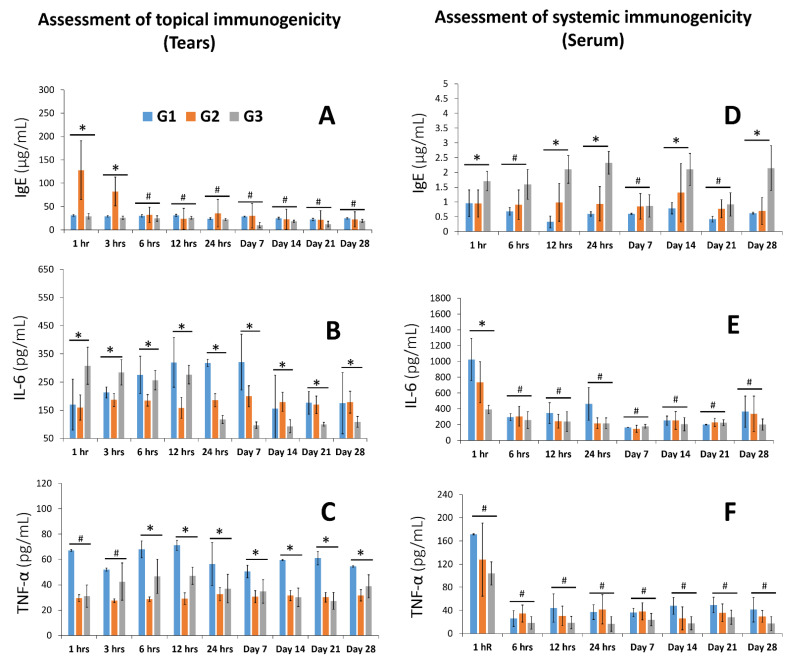
Serum and tear levels of IgE, IL-6, and TNF-α after treatment with En+/En− hLMSCs in rabbits. (**A**–**C**) Bar charts displaying the ELISA-determined concentrations of the cytokines IgE, IL-6, and TNF-α in rabbit serum. (**D**–**F**) Quantitative analysis of rabbit tear samples for the cytokines interleukin (IL)-6, tumor necrosis factor α, and interleukin (IL)-1 presented as bar graphs. Both the experimental and control groups showed a downward trend in cytokine levels, suggesting that there was no localized toxicity in the eyes of the recipients. * *p* ≤ 0.05; # *p* > 0.05. G1—Sham treated group; G2—Treated with En− hLMSCs; G3—Treated with En+ hLMSCs.

**Figure 8 cells-12-00876-f008:**
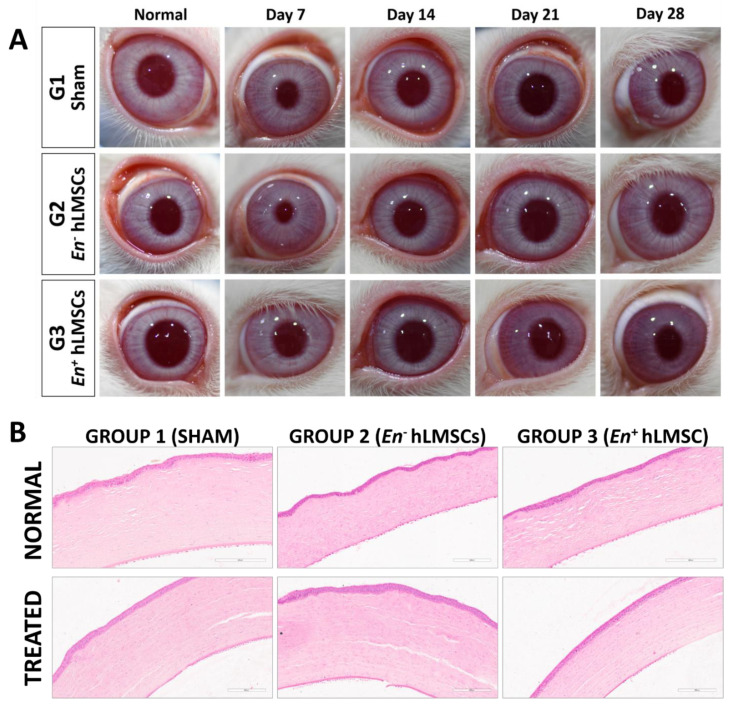
Rabbit eyes and corneal histopathology after En+/En− hLMSC treatment. (**A**) Clinical photographs of normal and injured rabbit eyes taken over a 28-day period showed no evidence of inflammation or irritation in the injured eyes. The Nikon D7200 and Nikon AF-S VR Micro-NIKKOR 105 mm f/2.8 G IF-ED lens were used to take the images. (**B**) Histopathological sections of normal and treated corneas represented by a panel of representative photomicrographs. Magnification: 40×; Scale: 200 µM. Group 1 received no treatment (sham group); Group 2 received En−hLMSCs; and Group 3 received En+ hLMSCs.

## Data Availability

Data are available on request due to institutional policies.

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
