# Peer review of "Pre-Clinical Evaluation of Efficacy and Safety of Human Limbus-Derived Stromal/Mesenchymal Stem Cells with and without Alginate Encapsulation for Future Clinical Applications"

_cells, 2023, doi:10.3390/cells12060876_

Round 1

Reviewer 1 Report

The article is very meaningful. However, some points still need to be further improved.

1 In the mouse injury model, the degree of injury should be consistent in all experimental animals. On Page 3, line 124-125, the authors said Central epitheium and anterior stroma were debrided in the right eye of the mice, and on Page 6, line 266-267: “Algerbrush II (Accutome Inc., Pennsylvania, USA) with a 0.5 mm burr was used to gently rotate the right eye's central cornea in a circular motion for 15-20 seconds. What is the thickness of the anterior corneal stroma removed by the author? What method is used to ensure the same thickness of the removed corneal stroma?

2 Page 9, line 355-358: Statistical methods should be described in some more detail. For example, which data was used for t-test, and which data was used for ANOVA testing?

3 How to evaluate the degree of scar?

In order to better explain the effect of drugs, it is necessary to compare their effect on the scar, so it is best to show the area and depth of the scar. The author used AS-OCT to compare the scars. Can authors give the relevant data of the scars? In addition, corneal confocal microscopy may better reflect the healing and scarring.

Reviewer 2 Report

The article entitled "Pre-clinical evaluation of efficacy and safety of human limbus-derived stromal /mesenchymal stem cells with and without alginate encapsulation for future clinical applications"  by Damala et al is an interesting and well-conducted research on MSC for the treatment of corneal wounds. I have not any specific criticism on the subject matter of this manuscript or on its quality.  On the other hand It's too long , it must be shortened by at least 40 %, moving a large amount  of the experimental findings in the section of supplementary data. 

Round 2

Reviewer 1 Report

The paper is well organized and I suggest it be accepted.

However, the figures should be improved. For example, the authors need to add a ruler or indicate its magnification in Figures 2,3 and 8